# Make Privacy Renewable! Generating Privacy-Preserving Faces Supporting Cancelable Biometric Recognition

Tao Wang[†]
Nanjing University of Aeronautics
and Astronautics
Nanjing, China
wangtao21@nuaa.edu.cn

Yushu Zhang[*†]
Nanjing University of Aeronautics
and Astronautics
Nanjing, China
yushu@nuaa.edu.cn

Xiangli Xiao
Nanjing University of Aeronautics
and Astronautics
Nanjing, China
xiaoxiangli@nuaa.edu.cn

Lin Yuan
Chongqing University of Posts and
Telecommunications,
Chongqing, China
yuanlin@cqupt.edu.cn

Zhihua Xia
Jinan University
Guangzhou, China
xia_zhihua@163.com

Jian Weng
Jinan University
Guangzhou, China
cryptjweng@gmai.com

## ABSTRACT

The significant advancement in face recognition drives face privacy protection into a prominent research direction. Unlike de-identification, a recent class of face privacy protection schemes preserves identifiable formation for face recognition. However, these schemes fail to support the revocation of the leaked identity, causing attackers to potentially identify individuals and then pose security threats. In this paper, we explore the possibility of generating privacy-preserving faces (not features) supporting cancelable biometric recognition. Specifically, we propose a cancelable face generator (CanFG), which removes the physical identity for privacy protection and embeds the virtual identity for face recognition. Particularly, when leaked, the virtual identity can be revoked and renew as another one, preventing re-identification from attackers. Benefiting from the designed distance-preserving identity transformation, CanFG can guarantee separability and preserve recognizability of virtual identities. Moreover, to make CanFG lightweight, we design a simple but effective training strategy, which allows CanFG to require only one (rather than two) network for achieving stable multi-objective learning. Extensive experimental results and sufficient security analyses demonstrate the ability of CanFG to effectively protect physical identity and support cancelable biometric recognition. Our code is available at https://github.com/daizigege/CanFG.

[*]Corresponding author

[†]Also with  Guangdong Provincial Key Laboratory of Novel Security Intelligence Technologies.

## CCS CONCEPTS

• **Security and privacy** → **Privacy protections**; **Usability in security and privacy**; • **Computing methodologies** → *Biometrics*.

## KEYWORDS

Face privacy; cancelable biometrics; virtual identity

**ACM Reference Format:**

Tao Wang, Yushu Zhang, Xiangli Xiao, Lin Yuan, Zhihua Xia, and Jian Weng. 2024. Make Privacy Renewable! Generating Privacy-Preserving Faces Supporting Cancelable Biometric Recognition. In *Proceedings of the 32nd ACM International Conference on Multimedia (MM '24), October 28-November 1, 2024, Melbourne, VIC, Australia.* ACM, New York, NY, USA, 9 pages. https://doi.org/10.1145/3664647.3680704

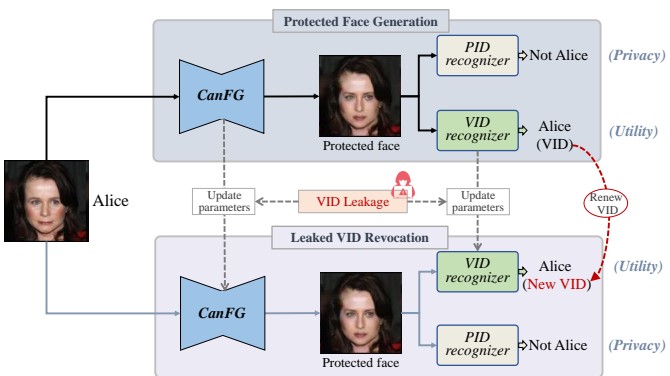

**Figure 1: Illustrative example of the proposed CanFG. Regarding the protected face of Alice, the physical identity (PID) recognizer cannot determine it as Alice, but the virtual identity (VID) recognizer can. Furthermore, when compromised, the used VID can be revoked and renewed as another VID.**

## 1 INTRODUCTION

Face recognition has made significant progress in recent years, and is widely applied in various fields, e.g., intelligent surveillance

**Table 1: Main advantages of CanFG over mainstream schemes, where PID means physical identity.**

|  | PID-protected | Attribute-retained | Cancelable |
|---|:---:|:---:|:---:|
| Li *et al.*[13] | ✗ | ✓ | ✗ |
| PRO-Face[36] | ✗ | ✓ | ✗ |
| IVFG[37] | ✓ | ✗ | ✗ |
| CanFG | ✓ | ✓ | ✓ |

and electronic payment. Nevertheless, the large-scale and non-consensual collection of face images also raises concerns about individual privacy. In 2020, Clearview AI experienced an unauthorized intrusion involving 3 billion face images. Because of the immutable nature of faces, malicious disclosure or misuse can result in irreversible harm to individuals.

To protect face privacy, many countries and international organizations issued relevant laws and regulations, e.g., the general data protection regulation (GDPR). Researchers also devotes considerable effort to designing schemes for enhancing face privacy, primarily focusing on generative model-based anonymization [2–4, 10, 14, 21, 33, 35] and adversarial perturbation-based obfuscation [1, 7, 25, 26, 34]. However, almost all schemes aim to remove identifiable information from faces (known as de-identification), thus preventing recognition via machine vision. As a result, face recognition fail to operate properly, greatly diminishing the accuracy and convenience of identity verification.

To address the above issue, other schemes preserve identity features for face recognition while anonymizing the facial visual appearance. Li *et al.* [13] designed a face anonymization model, which can adaptively modify identity-independent attributes to obfuscate visual appearance. Unfortunately, as the number of altered attributes increases, the loss of identity information becomes severe, leading to a significant decrease in recognizability. PRO-Face [36] leverages the idea of data hiding to hide identity information into the obfuscated face. In this way, the protected face is visually similar to the obfuscated one, but the machine vision can distinguish its original identity. The aforementioned schemes rely on *physical identity (PID) for recognition, which can be extracted from faces in the real physical world*. Nevertheless, PID is unique and immutable (unrevocable), and would be exposed forever once leaked. Attackers can pose permanent security threats to individuals based on the leaked PID, e.g., identity impersonations and reputation violations.

Compared to PID, virtual identity (VID) provides stronger security for face recognition. Representatively, IVFG [37] generates faces with different VIDs via key control, where the same PID corresponds to the same VID. IVFG provides satisfactory identifiability and privacy protection, but cannot retain attributes unrelated to identity, thus disabling many vision tasks, e.g., pose detection and background recognition. Additionally, it brings the additional problem of key management and lacks the theoretical guarantee for separability. More importantly, IVFG still fails to support cancelability. *Because the VID remains available with the corresponding key, when it should be revoked due to leakages.* However, since remembering the leaked VIDs accurately is difficult, the user may mistakenly utilize the leaked VIDs. In such a case, the attacker still

can identify the user for potential threatening behaviors such as malicious tracking and privacy invasion.

In this paper, we present CanFG, which can generate privacy-preserving faces supporting cancelable biometric recognition. Unlike the works of cancelable biometrics [9, 12] which output protected identity features, CanFG focuses on generating protected faces. Fig. 1 illustrates the usage of CanFG. After Alice's face image is protected by CanFG, the PID recognizer cannot determine it as Alice, but the VID recognizer can. In this way, Alice's privacy is protected while face recognition utility is preserved. Furthermore, when the used VID is leaked, it can be revoked and renewed as another VID, avoiding the problem of identity leakage for life. Since the identifiable information (i.e., VID) has been renewed, the attacker can no longer identify users based on this leaked VID, avoiding further security threats. Table 1 shows the main advantages of CanFG over the above schemes.

We summarize the main contributions as follows:

- We propose a privacy-preserving face generator, i.e., CanFG, which can support cancelable biometric recognition while protecting physical identity.
- We construct a distance-preserving identity transformation based on orthogonal matrices, which can ensure separability and maintain recognition performance of virtual identities.
- We design a simple but effective training strategy inspired by data hiding, which allows CanFG to require only one (rather than two) network for achieving physical identity removal and virtual identity embedding simultaneously.

## 2 RELATED WORK

### 2.1 Identifiable Face Privacy Protection

Identifiable face privacy protection means protecting face privacy while allowing face recognition. Privacy-preserving face recognition schemes remove all available content except for identity. Wang *et al.* [29] discovered that human vision primarily relies on low-frequency information for image recognition, whereas machine vision focuses on both low and high-frequency information. Building upon this, many schemes [16, 17, 32] remove identity-irrelevant low-frequency features in the frequency domain, ensuring that protected images are used only for face recognition. However, the protected results generated by these schemes lose a significant amount of available information, and thus cannot support simple computer vision tasks, such as face detection.

To preserve more utility, recent schemes only change private content while preserving identity features in faces. For private attributes, some schemes [18, 38] are built on attribute manipulation frameworks to invert privacy attributes while maintaining as much identity performance as possible. For private visual identity, some schemes [13, 30, 36] anonymize the face appearance to prevent identity perception by human vision, but retain identity features to allow recognition by machine vision.

All of the above works adopt the physical identity (PID) for recognition. However, the PID is unique for anyone and cannot be revoked once compromised. Therefore, recent schemes [31] consider using the virtual identity (VID) to facilitate face recognition. IVFG [37] generates faces with the same VID for different faces with the same PID, where the VID is controlled by the user key. However, this

scheme fails to preserve attributes except identity. Furthermore, its simple implementation of separability through inter-group classification loss lacks theoretical guarantees. While Anonym-Recognizer [20] solves the limitations of IVFG, the used binary relationship cyphertext lacks robustness in recognition. Because once one bit goes wrong, the recognition result goes wrong. Moreover, it utilizes two networks for anonymization and recognition, which is not friendly to applications with constrained resources.

## 2.2 Cancelable Biometrics

In biometrics, the unique biometric template can accurately identify individuals but pose significant security risks if compromised. Cancelable biometrics [15, 19, 28] transform the unique biometric template into different templates for identification, enhancing security for biometrics. Specifically, cancellable biometrics must satisfy four characteristics: 1) *Non-invertibility*: If the protected template is compromised, it is difficult to recover the original biometric sample. 2) *Comparable performance*: Recognition performances before and after protection are comparable. 3) *Diversity:* The same biometric sample can be protected into multiple templates. 4) *Revocability*: If the protected template is compromised, it can be revocable and replaced with a new template.

Several cancelable biometrics schemes have been proposed and follow the following process: 1) During enrollment, a cancelable template is transformed by some designed transformations for the features extracted from the biometric sample, and then registered in the database. 2) During authentication, the cancelable template is created for the input biometric sample in the same way, and identification is performed by comparing it with the cancelable templates stored in the database. Therefore, the main point is to how to construct the transformation, including filtration [24], deformation [22], and encryption [9]. Interestingly, some recent schemes [5, 12] employ data hiding to achieve cancelable biometrics. They guarantee the non-invertibility of the biometric sample by hiding the transformed template into the cover image, because only the specific recognizer can access the transformed template.

## 3 THE PROPOSED CANFG

### 3.1 Overview

We present a cancelable face generator (CanFG), which can support cancelable biometric recognition while protecting face privacy. As shown in Fig. 1, given an original face image, the proposed CanFG can generate a protected face image. Firstly, the protected face belongs to an anonymized physical identity (PID), which prevents unauthorized extraction of the real PID, thus protecting face privacy. Secondly, the protected face also is embedded with a virtual identity (VID), which can be extracted by a specific VID extractor and can be revoked when leaked, thus supporting cancelable recognition.

As shown in Fig. 2, the process of utilizing CanFG follows the standard cancelable biometric recognition. In the enrollment, the registered face is protected by CanFG and then stored in the database. In the authentication, the input face is also protected by CanFG, and then the VID extractor obtains the VIDs of the input face and the registered face for verification.

Before describing how to train the CanFG, we introduce a new identity transformation and a new physical identity remover.

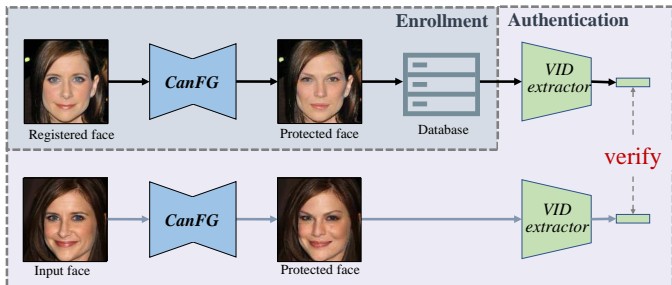

**Figure 2: Cancelable face recognition via CanFG.**

### 3.2 Distance-preserving Transformation

In this subsection, we aim to construct a distance-preserving transformation to transform PIDs into VIDs, i.e., neighboring points in the physical space remain at the same distance in the virtual space after the transformation. There are two main advantages for demanding distance preservation:

- Ensuring separability: Separability can prevent different PIDs from being mapped to the same VID, which can avoid recognition errors. IVFG [37] adopts loss optimization to try to achieve separability, but fails to guarantee it theoretically.
- Maintaining performance: Since the identity distance can be preserved, the recognition performance in the VID space is fully equivalent to that in the PID space.

We select the orthogonal transformation as the distance-preserving transformation. The orthogonal transformation are equidistant and equiangular, thus conforming to the invariance of the Euclidean and cosine distances of the identity features. For any pair of face images, the extracted identity features ($\mathbf{e}_1, \mathbf{e}_2$) from them remain the same Euclidean and cosine distances after the orthogonal transformation:

$$Dis(\mathbf{Q} \cdot \mathbf{e}_1, \mathbf{Q} \cdot \mathbf{e}_2) = Dis(\mathbf{e}_1, \mathbf{e}_2), \tag{1}$$

where $\mathbf{Q}$ is an orthogonal matrix, $Dis(\cdot, \cdot)$ can be either Euclidean distance or cosine distance. In addition, orthogonal matrices are not unitary. Security for identity transformations can be enhanced by generating random orthogonal matrices. Specifically, different face recognition systems use different orthogonal matrices $\{\mathbf{Q}_1, \mathbf{Q}_2, ...\}$ so that obtaining different VIDs, thus preventing data-linking attacks between them. That can be formulated as:

$$\mathbf{Q}_1 \cdot \mathbf{e}_1 \neq \mathbf{Q}_2 \cdot \mathbf{e}_1. \tag{2}$$

### 3.3 Auxiliary Physical Identity Remover

In this subsection, we construct a PID remover for assisting privacy protection. Its backbone is set as U-Net [23]. In order to train the PID remover, a pre-trained PID extractor $E_{pid}$ (ArcFace [6]) is used to facilitate the adversarial training, which guides the variation of PID through the identity deviation loss:

$$\mathcal{L}_{pid} = cos(E_{pid}(X_i), E_{pid}(X_a)), \tag{3}$$

where $X_i$ is the original face, $X_a$ is the anonymized face, and $cos(\cdot, \cdot)$ represents the cosine similarity.

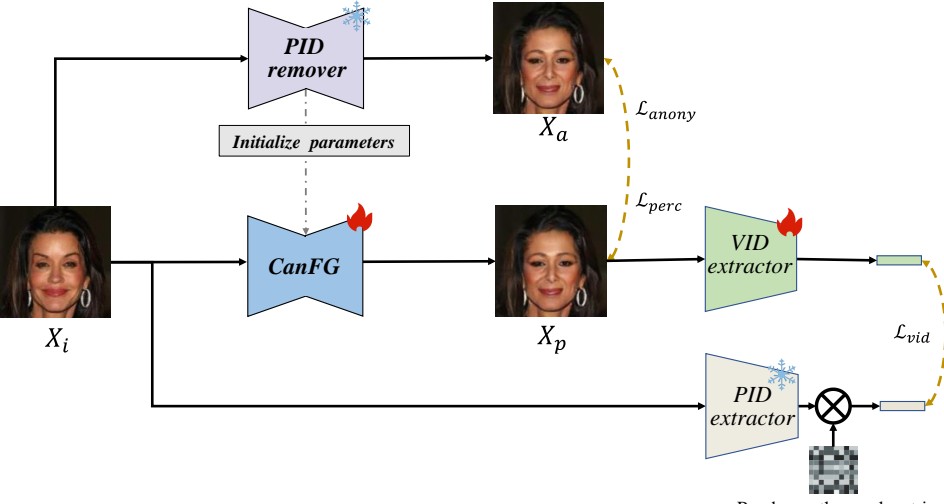

**Figure 3: The training of CanFG. Inspired by data hiding, we consider the anonymized face $X_a$ as the cover image, and the transformed virtual identity as the embedded information. Moreover, we initialized CanFG with the parameters of PID remover.**

However, this identity deviation does not specify a specific optimization direction and thus can easily lead to optimization difficulties. In addition, the uncontrolled protected face may map to other person's identity, causing disturbance to him or her. Therefore, we consider adopting the mean PID features $\overline{pid}$ of all faces as the direction of optimization, and generate anonymized faces by moving away from the original PID and approaching the mean PID:

$$\mathcal{L}_{pid} = cos(E_{pid}(X_i), E_{pid}(X_a)) - cos(E_{pid}(X_a), \overline{pid}). \quad (4)$$

In order to reduce the alteration of identity-unrelated attributes, we use the reconstruction loss to retain more usable information:

$$\mathcal{L}_{rec} = \|X_i - X_a\|_1. \quad (5)$$

Moreover, adversarial learning is utilized to enhance the clear visual quality of the anonymized results. In particular, we employ WGAN-GP [39] to steady the training process. The adversarial loss is expressed as $\mathcal{L}_{adv} = \mathcal{L}_{adv}^P + \mathcal{L}_{adv}^D$, where the loss of the remover and the discriminator $D$ can be expressed as:

$$\mathcal{L}_{adv}^P = -D(X_a) \quad (6)$$

and

$$\mathcal{L}_{adv}^D = -D(X_i) + D(X_a) + \lambda gp(\|\nabla_{X'}D(X_a)\|_2 - 1)^2 \quad (7)$$

Finally, the objective for the PID remover is formulated as:

$$\mathcal{L}_{total} = \mathcal{L}_{adv} + \lambda_1 \mathcal{L}_{pid} + \lambda_2 \mathcal{L}_{rec} \quad (8)$$

where $\lambda_i$ is the hyperparameter for balancing losses.

### 3.4 A Novel Training Strategy for CanFG

**Training difficulty.** The CanFG can not only protect the physical identity via PID removal, but also support cancelable recognition via VID embedding. An intuitive idea to train CanFG is to first perform PID removal followed by VID embedding. Such idea has been validated in PRO-Face [36] and Anonym-Recognizer [20], but requires two networks to form the CanFG. To minimize resource

consumption, a network can be used as the backbone of CanFG for training, but it is difficult for the CanFG to converge well with the multi-task objective, leading to a not satisfactory performance.

To solve the above difficulty, we design a simple but effective training strategy inspired by data hiding. Data hiding is a technique for embedding information in digital media without affecting obvious changes. In the case of image media, data hiding can embed information (like text) as a secret into an image (called the cover image), without affecting the appearance of the cover image. Fig. 3 illustrates the training of CanFG. With reference to data hiding, we consider the anonymized face $X_a$ obtained by PID remover as the cover image, and the transformed virtual identity as the embedded information. In this way, the protected image $X_p$ will have the same appearance as the anonymized face $X_a$ while being embedded with the virtual identity. Specifically, the training strategy contains the learning objective and a training trick.

*3.4.1 Data hiding-based learning objective.* We state the learning objective in terms of PID removal and VID embedding.

**Physical identity removal.** Firstly, the protected face should have a different physical identity from the original face, thus preventing recognition by unauthorized face recognizers. For this purpose, we set up a pixel-level loss to reduce the difference between the protected face and the face anonymized by the PID remover:

$$\mathcal{L}_{anony} = \|X_p - X_a\|_1. \quad (9)$$

Furthermore, perceptual loss is employed to draw close to the protected face and anonymized face features to enhance the perceptual similarity between $X_p$ and $X_a$:

$$\mathcal{L}_{perc} = \|VGG(X_p) - VGG(X_a)\|_1, \quad (10)$$

where $VGG(\cdot)$ is the pretrained VGG-based deep feature extractor.

**Virtual identity embedding.** Secondly, we require a VID extractor $E_{vid}$ to recognize the virtual identity in the protected face.

Specifically, we employ the Euclidean distance to reduce the distances between features:

$$\mathcal{L}_{vid} = \left\| E_{vid}(X_p) - \mathbf{Q} \cdot E_{pid}(X_a) \right\|_2, \qquad (11)$$

where the orthogonal matrix $\mathbf{Q}$ is random but fixed in training, and should be discarded after training. For different application systems, the orthogonal matrices they adopted are random and therefore difficult to duplicate. If the VID leaks, an arbitrary system can replace with another random matrix to renew the VID. Similar to PID protector, adversarial learning is used to enhance quality.

Finally, the objective for CanFG and VID extractor is formulated as follows:

$$\mathcal{L}_{total} = \mathcal{L}_{adv} + \lambda_3 \mathcal{L}_{anony} + \lambda_4 \mathcal{L}_{perc} + \lambda_5 \mathcal{L}_{vid} \qquad (12)$$

where $\lambda_i$ is the hyperparameter for balancing losses.

*3.4.2 Fine tuning-based training trick.* Physical identity removal and virtual identity embedding are intuitively non-conflicting during training. However, in the practical training process, the lack of stability in the training makes it difficult for CanFG to converge. To this end, we introduce a fine tuning-based training trick. Specifically, before training CanFG, the parameters of the PID remover need to be initialized to the CanFG so that it has the ability to generate anonymized faces. In this way, the network structures of CanFG and PID remover need to be consistent, both being U-Net.

## 3.5 Analysis of Cancelable Recognition

In this section, we will analyze the reason that why CanFG supports the cancelable biometric recognition.

*3.5.1 Non-invertibility.* When the protected face is compromised, the attacker can only obtain an anonymized (not real) PID. Such false PID have no obvious correlation with the real PID, so it is difficult for an attacker to recover the real one. More irreversible analysis can be found in the *security analysis* 4.5. Therefore, CanFG satisfies the requirement of non-invertibility.

*3.5.2 Comparable Performance.* Since the orthogonal transformation is equidistant and equiangular, the VID space maintains the same distance as the PID space via Euclidean or cosine distances. Therefore, the recognition performance is theoretically completely unchanged before and after the transformation. Since the embedding and extraction of VIDs loses some feature information, making the recognition performance biased but still comparable to the original performance. Therefore, CanFG satisfies the requirement of comparable performance.

*3.5.3 Diversity.* Since orthogonal matrices are not unique and are numerous in quantity, the same PID can be transformed into different VIDs for identification. Therefore, CanFG satisfies the requirement of diversity.

*3.5.4 Revocability.* When VIDs are leaked mainly because of the unsafe protection of VID extractor, the attacker is able to identify individuals based on the leaked VID. To prevent it, we can update the parameters of CanFG and VID extractor by changing the orthogonal matrix, and then renew VIDs while disabling the leaked VIDs, preventing re-identification from attackers. Therefore, CanFG satisfies the requirement of revocability.

## 4 EXPERIMENTAL RESULTS

### 4.1 Setup

*4.1.1 Dataset.* **CelebA** is a widely used face dataset, which contains 202,599 face images of 10,177 celebrity identities. We utilized 180,000 images sorted by indices as training data, and the remained images as testing data. **VGGFace2** is a large-scale face dataset, which contains over 3.31 million images of 9,131 identities. To verify the ability of CanFG to generalize to other datasets, VGGFace2 is only used for testing.

*4.1.2 Implementation Details.* Firstly, the pretained PID remover is optimized with Adam optimizer with $\beta_1 = 0.5, \beta_2 = 0.99$. The batch size is set to 128, the initial learning rate is set to 0.0002 and the weighting hyperparameter is set to $\lambda_1 = 10, \lambda_2 = 100$. Secondly, we initialize the parameters of CanFG to the trained parameters of the PID remover. CanFG is also optimized by Adam optimizer with $\beta_1 = 0.5$ and $\beta_2 = 0.99$. The batch size is set to 64, the initial learning rate is set to 0.0002, and the weighting hyperparameters are set to $\lambda_3 = 100, \lambda_4 = 10, \lambda_5 = 500$.

*4.1.3 Baselines.* To the best of our knowledge, CanFG is the first model that supports cancelable biometric recognition for privacy-preserving faces. Some VID-based schemes are similar to CanFG, but fail to satisfy cancelability. For this reason, we choose a representative VID-based scheme (i.e., IVFG [37]) and a representative PID-based scheme (i.e., PRO-Face [36]) as references to show that the performance of CanFG is comparable. For IVFG, since the training details are not published, we just used the results from the original paper. For PRO-Face, we choose FaceShifter as the obfuscator and select a random face as the target.

### 4.2 Evaluation on Privacy Protection

The privacy protection goal of CanFG is to prevent the machine recognition of the original physical identity.

**Qualitatively,** Fig. 4 illustrates the protected results generated by CanFG. It can be observed roughly that the protected face has a high visual similarity with the original one, which is contributed by the fact that CanFG modifies mainly the identity-related regions and preserves more irrelevant attributes, e.g., the background and hair. Therefore, compared to IVFG [37], which changes all image regions, CanFG is more applicable to practical application scenarios such as surveillance video. **Upon further scrutiny**, we can find some obvious appearance changes in facial areas of the protected face, including the eyes, nose, and mouth. In this way, it is highly probable to visually consider that the protected face and the original face do not belong to the same physical identity. Therefore, the face privacy can be protected well in qualitative analysis.

**Quantitatively,** we utilize an advanced face recognition API (Face++) to evaluate the protection performance. We perform protection on one of the paired faces and later evaluate the identity

**Table 2: PID protection success rates under different matching thresholds via Face++.**

|  | PSR@$\theta$=62.3% | PSR@$\theta$=69.1% | PSR@$\theta$=74.0% |
|---|---|---|---|
| CelebA | 90.4% | 97.0% | 99.2% |
| VGGFace2 | 84.9% | 94.3% | 98.2% |

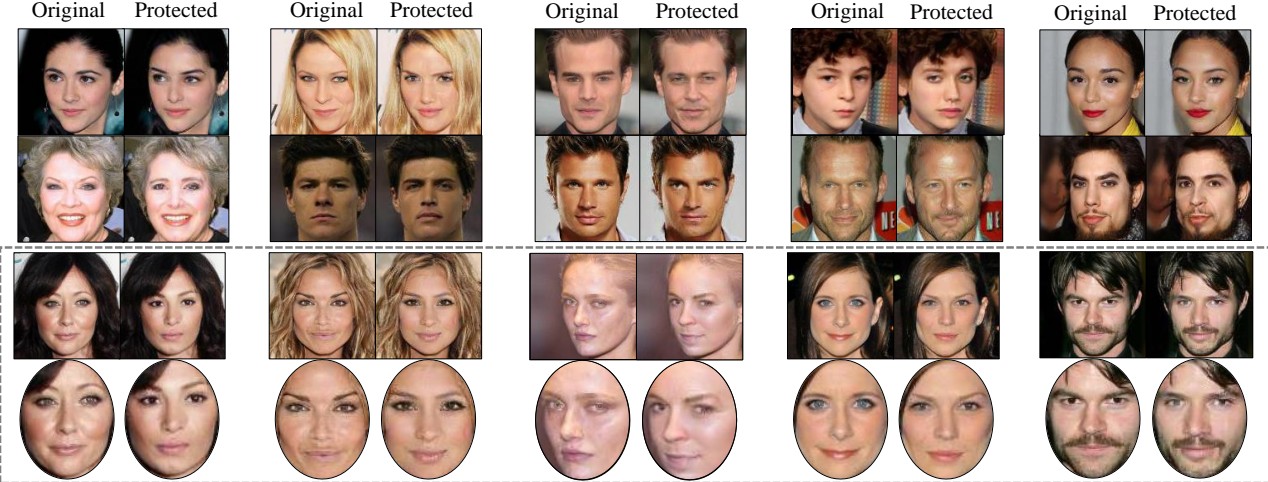

**Figure 4: The visual results of protected faces by CanFG. The facial areas of the last row protected images are enlarged for scrutiny of identity changes.**

similarity of the paired faces. To reveal the distribution properties of the similarity, Fig. 5 demonstrates the corresponding box-plots. It is clear that the CanFG obtains the average scores of less than 50% in CelebA and VGGFaces, which is significantly lower than the matching threshold ($\theta$=74.0%). Further, we calculated the protection success rate (PSR) at different matching thresholds, including 62.3%, 69.1%, and 74.0%. As shown in Table 2, CanFG achieves over 90% protection success rate on the CelebA, as well as a high protection rate on the VGGFace2, which is not seen in training. It should be note that adjusting $\lambda_1$ yields a higher PSR but also reduces the degree of attribute retention. Therefore, the face privacy can be protected well in quantitative analysis.

### 4.3 Evaluation on Recognizability Preservation

In this part, we evaluate the ability of CanFG to preserve recognizability. The orthogonal matrix in CanFG is similar to the key in IVFG [37], and they are both able to control the change of virtual

**Table 3: Recognition performance of IVFG, PRO-Face and CanFG on CelebA and VGGFace2.**

|  | CelebA | | VGGFace2 | |
|---|---|---|---|---|
|  | EER↓ | AUC↑ | EER↓ | AUC↑ |
| Original | 0.027 | 0.990 | 0.074 | 0.964 |
| IVFG[37] | 0.196 | 0.887 | - | - |
| PRO-Face[36] | 0.112 | 0.954 | 0.191 | 0.888 |
| CanFG | **0.045** | **0.988** | **0.101** | **0.951** |

identity. To be comparable, this experiment also requires IVFG to use the same key. Both IVFG and PRO-Face used pretrained Arcface [6] (SEResNet50) for identity extraction, and only CanFG employed a specialized VID extractor (SEResNet50), which guarantees that virtual identity will not be compromised easily.

Table 3 shows the recognition performance on protected faces by IVFG, PRO-Face, and CanFG, including equal error rate (EER) and area under curve (AUC). It can be seen that, CanFG achieves the best recognition performance, which only slightly reduces the original performance. This is mainly attributed to the distance-preserving identity transformation.

**Different PID extractors.** Different PID extractors can obtain different identity features, thus affecting the recognition performance. Therefore, we set the PID extractor as various models, including InceptionResNet [27], IResNet50 [8], SEResNet50 [11], and IResNet100 [8]. PRO-Face also supports different models, and thus we compare CanFG with it. Fig. 6 shows the ROC curves on CelebA and VGGFace2. As it can be seen, CanFG reaches AUC values very close to that of the original one in CelebA. Due to the domain discrepancy of the datasets, CanFG does not achieve the same excellent performance on VGGFace as it does on CelebA, but it still maintains high AUC values on all models. Compared to PRO-Face, CanFG also keeps higher AUC values on all models and datasets. Additionally,

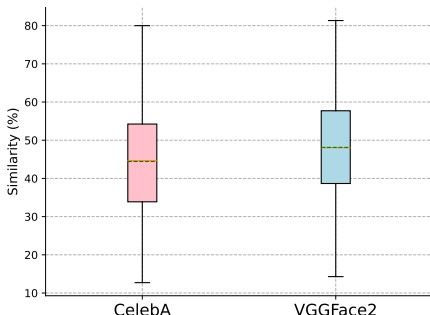

**Figure 5: PID similarities between the paired faces via Face++, one of which is protected by CanFG.**

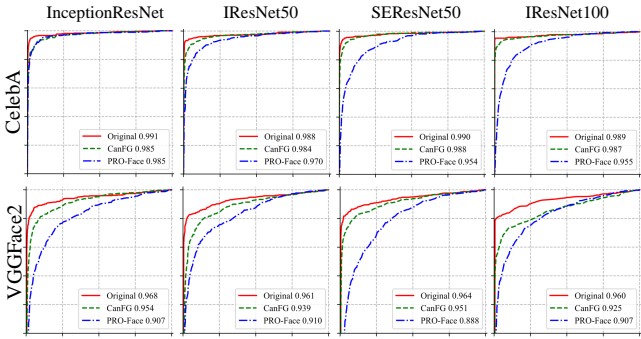

**Figure 6: ROC curves of four PID extractors.**

**Table 4: Recognition performance among different orthogonal matrices on CelebA.**

|  | $Q_1$ | | $Q_2$ | | $Q_3$ | |
|---|---|---|---|---|---|---|
|  | EER↓ | AUC↑ | EER↓ | AUC↑ | EER↓ | AUC↑ |
| $Q_1$ | **0.045** | **0.988** | 0.502 | 0.500 | 0.507 | 0.513 |
| $Q_2$ | 0.502 | 0.500 | **0.057** | **0.986** | 0.498 | 0.486 |
| $Q_3$ | 0.507 | 0.513 | 0.498 | 0.486 | **0.038** | **0.989** |

*supplementary materials* present the true acceptance rate (TAR) of the recognition results with two false acceptance rate (FAR) values (0.01 and 0.1). CanFG has superior TAR values to PRO-Face.

**Different orthogonal matrices.** By changing the orthogonal matrix used for identity transformation, protected faces are embedded with different VIDs. We randomly used three orthogonal matrices to train CanFG and test the recognition performance. As shown in Table 4, CanFG acquires an AUC value of more than 0.986 with different matrices in training, retaining the recognition performance to a higher degree. Moreover, for CanFGs obtained from different orthogonal matrices, they have extremely poor recognition performance in utilizing virtual identities among each other, which prevents data interoperability between different service providers.

### 4.4 Feature Visualization

To demonstrate the effectiveness of CanFG in an understandable way, we visualize the physical and virtual identity features. UMAP is a nonlinear dimensionality reduction technique capable of preserving the local and global Euclidean distance relationship, which can help us to display identity features in 2D space.

In Fig. 7 (a), we show the relevant features of 28 face images of the same identity, including the original PID features, the anonymized PID features, the expected transformed VID features, and the actual extracted VID features. It is clear from the observation that the anonymized PID are farther away from the original PID, and hence the PID can be protected well. In addition, the extracted VID nearly overlaps with the region where the transformed VID are located, so that satisfactory recognition performance can be retained.

In Fig. 7 (b), we visualize extracted VID features for three different PIDs, each of which has 28 face images. We observe that each PID is assigned to a different VID and that there is significant spacing

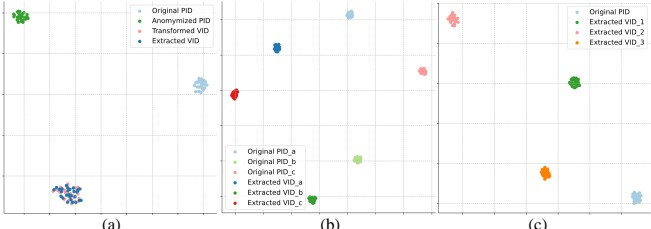

**Figure 7: Feature visualization via UMAP. (a) Related features corresponding to the same PID under the same Q. (b) VID features corresponding to different PIDs under the same Q. (c) VID features corresponding to the same PID under the different Q.**

between VIDs. In addition, we can clearly notice that the distance between VIDs approximates the distance between PIDs. This is due to the fact that UMAP is maintaining the Euclidean distances and is able to reflect the equidistant transformations performed in CanFG. In this way, the separability in virtual identities and the preservation of recognition performance can be theoretically guaranteed.

In Fig. 7 (c), we show the feature visualization for the same physical identity protected by different CanFGs. It can be noticed that the VIDs extracted by the corresponding VID extractors are separated from each other. This demonstrates that CanFG supports the availability of non-crossing VIDs for different service providers, avoiding data interoperability between service providers.

### 4.5 Security Analysis

The goal of the attacker is to obtain the real PID of the protected face. When the protected faces are leaked, because of the satisfactory visual naturalness, the attacker is tricked into thinking that the anonymized PID represented by the protected face is real. Therefore, the attacker has no will to try to recover the real PID. Even if the attacker has the will, it is difficult for him to deduce the true result since the anonymized PID does not have a direct correlation with the real PID. On the basis of the face leakage, we further assume that the attacker has access to additional information.

**Additional access to CanFG**. Since CanFG is irreversible, the attacker cannot recover the original face by CanFG directly. Of course, the attacker can train a reversible CanFG model by obtaining a large number of original and protected face pairs. To enhance security, we suggest adding conditional information to CanFG to generate diverse protected results.

**Additional access to VID extractor**. In this case, the attacker can extract the VIDs via the extractor but still obtain the anonymized PIDs. The failure to access CanFG makes it impossible to capture the correlation between PIDs and VIDs, and thus the real PIDs cannot be deduced. Considering that the VID is leaked, we can revoke it and replace with a new VID by updating the parameters of VID extractor, blocking attackers from using the leaked VID .

**Additional access to both CanFG and VID extractor.** In this case, the attacker can obtain the real PIDs and VIDs pairs. The matrix **Q** used in orthogonal transformation in CanFG is of 512 dimensions. Firstly, the attacker extracts 512 original PID features

**Table 5: Ablation experiment on CelebA**

|  | Utility | | Privacy | Visual quality | |
|---|---|---|---|---|---|
|  | EER↓ | AUC↑ | PSR↑ | FID↓ | SSIM↑ |
| Full strategy | 0.045 | 0.988 | 0.992 | 9.426 | 0.823 |
| W/o DH | 0.130 | 0.949 | 0.991 | 11.404 | 0.811 |
| W/o FT | 0.038 | 0.989 | 0.103 | 7.752 | 0.888 |
| W/o DH & FT | 0.112 | 0.953 | 0.984 | 10.996 | 0.816 |

**Table 6: Robustness test on CelebA.**

|  | EER↓ | AUC↑ |
|---|---|---|
| No process | 0.045 | 0.988 |
| Gaussian noise | 0.043(-0.002) | 0.988(-0.000) |
| JPEG compression | 0.107(+0.062) | 0.961(-0.027) |
| Median filtering | 0.123(+0.078) | 0.936(-0.052) |

of original faces to construct a matrix **Y**. Secondly, the attacker utilizes CanFG to generate protected faces and extracts 512 VID features via VID extractor to construct matrix **X**. Lastly, the value of orthogonal matrix **Q** can be estimated by $\mathbf{Q} = \mathbf{X}^{-1} \cdot \mathbf{Y}$. In this way, the attacker can use this matrix to estimate the real PID of the protected face. CanFG can replace the matrix **Q** with a new one by updating the parameters of CanFG and VID extractor, thus avoiding the continuous cracking of new protected faces. Of course, it is still crucial to manage the CanFG and VID extractor safely.

### 4.6 Ablation Experiment

We present ablation experiments to demonstrate the effectiveness of the training strategy. Specifically, the designed training strategy consists of the learning objective based on data hiding and the training trick based on fine-tuning. We discard the data hiding-based learning objective and instead incorporate the loss of the PID remover into the training, which is referred to as "w/o DH". In addition, we discard the fine tuning-based training trick, i.e., we do not initialize the parameters of the CanFG, which as "w/o FT".

Table 5 shows the performance of the ablation experiments, which include recognition performance (EER and AUC), protection performance (PSR), and visual quality (FID and SSIM). Without the data hiding-based learning objective (w/o DH), although the protection performance of PIDs is not affected, it makes the recognition performance and visual quality degrade. Without the fine tuning-based training trick (w/o FT), while the recognition performance and visual quality is improved, the protection performance of PIDs has declined sharply. Therefore, CanFG struggles to converge and fails to achieve superior performance on PID removal and VID embedding when lacking either of the above training strategies.

### 4.7 Additional Experiment

*4.7.1 Robustness test.* We used Gaussian noise, JPEG compression, and median filtering to evaluate the robustness of CanFG. According to Table 1, CanFG is least affected by Gaussian noise and most affected by median filtering. It is noteworthy that the AUC values are always above 90%, preserving an available recognition performance. Therefore, the robustness of CanFG is acceptable and also can be enhanced in the future.

*4.7.2 Distance preserving test.* Although the orthogonal transformation guarantees error-free distance preservation, the results extracted by the VID extractor are still in error from the expected VID. Therefore, we test the absolute errors between the PID distance of the paired faces before protection and the VID distance of the paired faces after protection. As shown in Fig. 8, both cosine and Euclidean distances of the PID space are well preserved in the VID space with low errors.

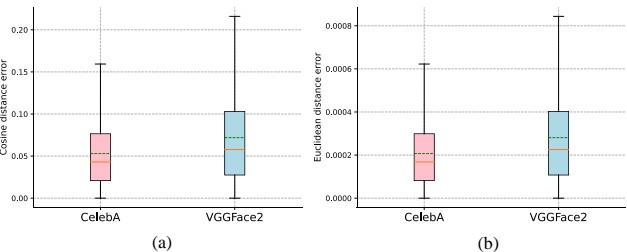

(a)  (b)

**Figure 8: The errors between PIDs distance and VIDs distance.**

*4.7.3 Real world faces test.* We randomly select face photos taken with cell phones in real world. As shown in Fig. 9, the cosine and Euclidean distances of the original PID still can be preserved to a high degree in the VID space. Since the real-world domain is different from the training data domain, the visual naturalness of the protected result is not perfect. In the future, we would like to abbreviate the intra-domain differences so that CanFG can be better applied to real scenarios.

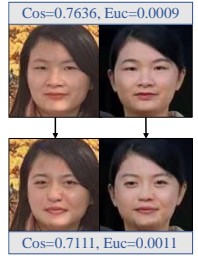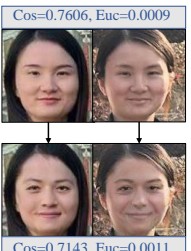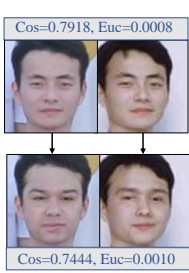

**Figure 9: Protected faces in the real world , where "cos" means the cosine distance and "Euc" means the Euclidean distance.**

## 5 CONCLUSION

In this paper, we propose CanFG, which can generate privacy-preserving faces supporting cancelable biometric recognition. Particularly, when leaked, the used virtual identity can be revoked and renewed as another one. We also introduced a distance-preserving identity transformation, which guarantees the separability and high recognizability of virtual identities. To stably train CanFG with a multi-objective loss, a new training strategy based on data hiding is designed. Sufficient experimental analyses and security analyses demonstrate the effectiveness of CanFG in protecting face privacy and maintaining high recognizability.

## ACKNOWLEDGMENTS

This work is supported in part by the National Key R&D Program of China under Grant number 2021YFB3100400, in part by the National Natural Science Foundation of China under grant numbers 62122032, U23B2023, and in part by the Guangdong Provincial Key Laboratory of Novel Security Intelligence Technologies under Grant number 2022B1212010005.

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
