# OpenReview forum: "Make Privacy Renewable! Generating Privacy-Preserving Faces  Supporting  Cancelable Biometric Recognition"
_acmmm.org/ACMMM/2024/Conference — MM2024 Poster_

### Official Review · Reviewer_b6kG · 2024-05-18

**Rating:** 4
**Confidence:** 3

**Summary:**

This paper introduces a cancelable face generator (CanFG), which removes the physical identity (PID) for privacy protection and embeds the virtual identity (VID) for face recognition. CanFG can revoke and renew the VID when it is leaked. Compared with existing schemes that rely on PID for recognition, CanFG uses VID to address the problem caused by the uniqueness of PID. More importantly, CanFG can update parameters to renew VID for cancelable recognition.

The method of CanFG consists of distance-preserving transformation (DPT) and an auxiliary physical identity remover (PIDR). In DPT, orthogonal transformation is adopted to map PID to VID, ensuring separability and maintaining the recognition performance of the PID space in the VID space. In PIDR, the anonymized face generated by PIDR is considered the cover image, and CanFG is initialized with the parameters of PIDR. The author considers that CanFG and PIDR have the same network structure in a fine-tuning-based training trick. The author also analyzes why CanFG supports cancelable biometric recognition in four aspects.  CanFG is evaluated on privacy protection and recognizability

**Strengths:**

This paper innovatively addresses the problem of achieving cancelable biometrics recognition while preserving privacy and provides a suitable solution. This work fills the gap in the joint analysis of revocability and privacy protection in face recognition.

The authors designed CanFG to generate protected faces, considering that current works on cancelable biometrics only output protected identity features while ignoring identity-independent attributes (e.g., hair, background), which can be used for other vision tasks. This approach fully considers the potential for practical application.

The authors use a fine-tuning trick to initialize the parameters of CanFG with those of the PID remover, which is beneficial for model training speed. CanFG can produce anonymous faces, making this training trick valuable when working with large amounts of image data.

In the experimental section, the authors conducted qualitative and quantitative analyses of privacy protection. They also replaced different modules (i.e., PID extractors, and orthogonal matrices) in CanFG to perform comparative experiments on recognizability preservation. The method of verifying the performance of the proposed technique is highly useful for reference.

**Limitations:**

1. In the orthogonal transformation of PID to VID, the Euclidean distance and cosine distance can be well kept unchanged. However, a simple linear transformation will make it easier for the attacker to crack.(See specific points in the justification section)
2. In the ablation experiment, the influence of different sub-training strategies on privacy
protection, recognition effect, and visual quality is analyzed. However, the lack of
experimental results based on the original model without any sub-strategies may lead to a less rigorous analysis.
3. In section 3.3, there is a lack of definitions of symbols and functions related to adversarial learning.
4. In section 3.4.1, there is no description of the definition of the VGG mapping when computing the perceptual loss.
5. The expression of the learning objective loss of Auxiliary Physical Identity(PID) Remover is rough, and the module structure of PID remover is not clear.
6. In this paper, some unique names (i.g., data hiding and cover image) lack the necessary explanation, which can easily cause misunderstanding and confusion.


Justification For Recommendation And Suggestions For Rebuttal:
1. In contrast to cacelable biometrics works which only output identity features, CanFG can generate protected faces. Although readers can get the why need to generate the full face instead of the identity features from the full-text understanding, I believe that pointing out the advantages of doing so when expressing this method is helpful for readers to understand this paper.
2. In section 3.3 ,the symbols and functions related to adversarial learning, i.g., $D(\cdot), \lambda, g, p$, need to be defined. Then,in section 3.4.1, the definition of $V G G(\cdot)$ should be given. It can reduce the confusion of readers.
3. In the PID remover, the learning objective loss is expressed as a combination of identity deviation loss, reconstruction loss and adversarial loss, but I'm a little confused about expressing it this way. If the author can add the structure diagram of PID remover in the paper, it will be more helpful to understand the definition of the learning objective loss and the workflow of PID remover.
4. The authors propose an effective training strategy for GANFGs inspired by data hiding. However, there is no brief explanation of the concept of data hiding in the full text, which makes me confused. in addition, there is no necessary explanation for the 'cover image' in line 441, which is easy to cause confusion. I hope the author can supplement the explanation.

5. In the transformation of PID to VID, orthogonal transformation has good invariance performance. However, since the linear transformation is easy to crack, a small bias can be added to enhance the security of the transformation with as little impact on the orthogonal characteristics as possible. The authors can consider this situation and implement comparative experiments.
6. In ablation experiments, authors could consider adding an experimental result based on the original model without adopting any training strategy. Based on this, the paper can better analyze the improvement ability of using different sub-training strategies for the original model.

Minor:
The Eq.(1) in line 324: $\left.\operatorname{Dis}\left(Q \cdot e_1, Q \cdot e_2\right)\right)=\operatorname{Dis}\left(e_1, e_2\right)$. There is an extra right parenthesis on the left side of the equation.

**Suitability:**

3

---

### Official Review · Reviewer_5FoC · 2024-05-21

**Rating:** 3
**Confidence:** 4

**Summary:**

This paper proposes CanFG. Its goal is to generate protected faces, carrying virtual identities, that can be recognized by a VID extractor and cannot be by any PID extractors. The authors employ a two-stage framework, first to train a PID remover that anonymizes the face, then to end-to-end train a pair of CanFG and VID extractor models to craft protected faces by embedding obfuscated IDs into the anonymized faces. The ID is revocable by opting for new CanFG and VID extractor models.

**Strengths:**

- Interesting research topic: Face protection is an important topic and is worth further exploration. Though cancelable biometric templates have been common in prior literature, the authors extend the goal to the creation of identity-cancelable faces, which is insightful.
- The proposed method seems to achieve its protection goal (with a premise of no further attacks) together with good performances.
- The experiments are extensive.

**Limitations:**

- I have concerns about some deliberate attacks. Though $X_i$ and $X_p$ are recognized by the PID extractor as different persons, (1) their visual appearances are still quite close as in Fig. 4, and (2) their relations are somehow "deterministic" (say, $X_p = E_{vid}^{-1}(Q\cdot E_{pid}(X_i))$, where $E_{vid}^{-1}$ is a fictional mapping and doesn't have to exist). My concern is, that a white-box attacker having access to the CanFG U-Net model can generate many $(X_i, X_p)$ pairs with a shared label $y$ (e.g., Alice). It is also within the attacker's capability to retrain a new (let's call it "PVID") extractor using both $\{X_i, y\}$ and $\{X_p, y\}$ as training data. The PVID extractor may incorporate both $(X_i, X_p)$ and recognize them as the same person. In short, retraining a new arbitrary extractor on both real and protected faces may nullify the virtual identity. This is my main concern. I invite the authors to consider this "retraining extractor" setting in further experiments.
- Similarly, the authors could also consider a generative attacker, that trains a U-Net to reversely fit $X_p$ to $X_i$.
- The effect of some loss terms (e.g., $\mathcal{L}_{adv}$) is unclear in the manuscript. The authors should consider further explaining them.
- The meaning of similarity in Fig. 5 is also unclear. Does it mean cosine similarity, the angle between two templates, or the percentage of angles? That can make a huge difference.
- The paper's writing can be improved.

I invite the authors to further elaborate on these issues. I am open to improving my score should my concerns be clarified.

**Suitability:**

3

---

### Official Review · Reviewer_nbdj · 2024-05-22

**Rating:** 4
**Confidence:** 4

**Summary:**

The paper presents a Cancelable Face Generator (CanFG) that generates privacy-preserving faces while supporting cancelable biometric recognition. CanFG removes physical identity for privacy protection and embeds virtual identity for recognition. The embedded virtual identity can be revoked and renewed if leaked, thus maintaining privacy and utility. The paper discusses the design and training of CanFG, as well as extensive experimental results and security analyses demonstrating its effectiveness.

**Strengths:**

1. **Robust Experimental Results**: The extensive experimental evaluations cover various aspects such as privacy protection, recognizability preservation, robustness, and real-world applicability. The results demonstrate that CanFG achieves high privacy protection and recognizability preservation while being robust against various attacks.
2. **Security Analysis**: The paper includes a detailed security analysis, demonstrating that CanFG meets the requirements of non-invertibility, comparable performance, diversity, and revocability. This comprehensive analysis strengthens the validity and reliability of the proposed method.

**Limitations:**

1. **Limited Dataset Diversity**: The experimental evaluations primarily focus on CelebA and VGGFace2 datasets. Evaluating CanFG on a broader range of datasets, e.g. FFHQ or CASIA-Webface, including those with more diverse demographics and environmental conditions, would enhance the generalizability of the results.
2. **Real-World Application Scenarios**: The authors could provide more detailed insights into how CanFG can be integrated into existing facial recognition systems and the potential challenges associated with its deployment in real-world scenarios.
3. **Implementation Complexity**: While the methodology is comprehensive, the implementation of CanFG may be complex and resource-intensive, especially for deployment in real-world scenarios with limited computational resources. Please list the comparison between CanFG and other methods in terms of the number of parameters and running time.
4. **Limited Experiments on Privacy Protection**: In Section 4.2, authors should give comparison of the state-of-art methods and their method on PID protection success rates. Moreover, the authors choose 3 threshold in the quantitative evaluation of PID protection success rates, please explain the reasons for choosing these thresholds.

**Suitability:**

3

---

### Official Review · Reviewer_jhk1 · 2024-05-25

**Rating:** 2
**Confidence:** 2

**Summary:**

The paper introduces a cancelable face generator (CanFG) that aims to achieve cancelable recognition, providing non-invertibility, comparable performance, diversity, and revocability. CanFG adopts a distance-preserving transformation to ensure separability and maintain performance, and an auxiliary physical identity remover for identity anonymization. The authors also design a training scheme to reduce resource requirements with a data hiding-based training strategy. The paper provides ample experiments and detailed analysis on the performance of CanFG, proving its effectiveness.

**Strengths:**

1. The proposed system has quite a solid theoretical ground. Analysis of each module, especially the distance-preserving transformation and the auxiliary physical identity remover provides an explicit and clear description of the idea.
2. The objective of minimizing resource consumption is commendable. By trying to avoid excess and heavy networks, the authors manage to control the size of the system with some tricks on the training strategy, which highly improves the practicability.
3. Abundant experiments on diverse setups prove the effectiveness of CanFG. They are well designed, covering many different aspects and therefore showing that CanFG empirically delivers on the promise of cancelable biometric recognition.

**Limitations:**

1. Writing can be further improved. Some sentences in the introduction, for example, could be reorganized and rearranged so that the key contributions of CanFG are more emphasized. For instance, it might be better if the narrative of the four requirements of a cancelable biometric scheme is more explicitly presented when the authors introduce existing studies, including PID ones and VID ones, since this seems to be the important concept throughout the whole work.
2. There should be some descriptions of the actual resources spent on training CanFG in Section 4 since the training strategy is intentionally designed to reduce resource consumption. It would be better if the authors compared the resource usage between CanFG and the baselines.
3. In equation (10), there seems to be a function VGG(·) without explanation.
4. The sentences in Lines 404 – 436 are a little confusing.
5. Some graphs are a bit difficult to read. The font size of Figs 5 – 8 is too small. Words in Fig 7 are extremely hard to read. Figs 1 – 4 are much better.

**Suitability:**

3

---

### Meta-Review · Area_Chair_SHUp · 2024-07-03

**Recommendation:** Accept (Poster)
**Confidence:** 5

**Metareview:**

This paper proposes an interesting face generation scheme, which produces faces with arbitrary, yet unique, identities which can be revoked and re-instatiated at the user's choice.
All reviewers agree on the strenghts and merits of the research work and outcomes. Many issues raised by the reviewers have been well addressed and fixed by the authors. At least one of the reviewers upgraded her/his score.
Apart form some minor concerns, most of the issues have been clarified and the paper can provide a valuable addition to the conference program.